# Linear Scalarization for Byzantine-robust learning on non-IID data

## Abstract

In this work we study the problem of Byzantine-robust learning when data among clients is heterogeneous. We focus on poisoning attacks targeting the convergence of SGD. Although this problem has received great attention; the main Byzantine defenses rely on the IID assumption causing them to fail when data distribution is non-IID even with no attack. We propose the use of Linear Scalarization (LS) as an enhancing method to enable current defenses to circumvent Byzantine attacks in the non-IID setting. The LS method is based on the incorporation of a trade-off vector that penalizes the suspected malicious clients. Empirical analysis corroborates that the proposed LS variants are viable in the IID setting. For mild to strong non-IID data splits, LS is either comparable or outperforming current approaches under state-of-the-art Byzantine attack scenarios.

## 1 Introduction

Most real-world applications using learning algorithms are moving towards distributed computation either: (i) Due to some applications being inherently distributed, Federated Learning (FL) for instance, (ii) or to speed up computation and benefit from hardware parallelization. We especially resort to distributing Stochastic Gradient Descent (SGD) to alleviate the heavy computation underlying gradient updates during the training phase. Especially with the high dimensionality of large-scale deep learning models and the exponential growth in user-generated data.

However, distributing computation comes at the cost of introducing challenges related to consensus and fault tolerance. In other words, the nodes composing the distributed system need to reach consensus regarding the gradient update. In the honest setting this can be done simply by a parameter server that takes in charge the aggregation of computation from the workers. However, machines are prone to hardware failure (crush/stop) or arbitrary behavior due to bugs or malicious users. The latter is more concerning as machines may collude and lead to convergence to ineffective models.

Since deep learning pipelines are involved in decision making at a critical level (e.g., computer-aided diagnosis, airport security...); it is crucial to ensure their robustness. We study robustness in the sense of granting resilience against malicious adversaries. More precisely, poisoning attacks that target the convergence of $SGD$. The adversarial model follows the Byzantine abstraction Lamport et al. (1982). The basis of distributed Byzantine attacks is the disruption of $SGD$'s convergence by tampering with the direction of the descent or magnitude of the updates. The robustness problem is highly examined and there exists a plethora of aggregations Blanchard et al. (2017); Alistarh et al. (2018); Yin et al. (2018); Damaskinos et al. (2018); Boussetta et al. (2021); El Mhamdi et al. (2018). Nonetheless, recent works Karimireddy et al. (2022; 2021) highlight the inability of these algorithms to learn on non-IID data. Indeed, defending against the Byzantine in a heterogeneous setting is not trivial. Naturally, aggregations rely on the similarity between honest workers to defend against the Byzantine. However, as data becomes unbalanced distinguishing malicious workers from honest ones becomes increasingly challenging: an honest worker may slightly deviate from its peers due to skewed data distribution.

Another weakness of current work is that most aggregation discard a subset of information either by dropping full gradient vectors or by eliminating a set of coordinates along each dimension of the submitted vectors. Nevertheless, dropping users' updates leads to (i) Degradation of the model's final accuracy, especially when no Byzantine attackers are present. (ii) It may also discard minorities with vastly diverging views as elaborated in Mhamdi et al. (2021).

Motivated by the latter and the scarcity of work on Byzantine Non-IID defenses, we study the Byzantine resilience of SGD on non-IID data. These two problems: (1) learning on non-IID data and (2) defending against Byzantine attacks are similar in the sense that in both cases the central server in charge of aggregating clients' updates is faced with conflicting information. In the Byzantine case the conflict stems from the discrepancy between the updates submitted by the malicious users and honest ones. Whereas, in the non-IID case it is a result of the difference in the data distribution between the users. In a realistic set up such as $FL$, the central server is faced with both challenges.

From an optimization perspective, the goal is to find the parameters of a model that simultaneously minimizes the loss associated with the local data of each client $f_i$. More precisely, the objectives in these cases maybe conflicting.

This type of optimization problem falls under multi-objective and can be formally expressed as follows:

$$\min_{\theta \in \mathbb{R}^d} \left( [f_1(\theta), f_2(\theta), ..., f_n(\theta)]^T \right). \tag{1}$$

Multi-Objective problems can be solved as mono-objective through linear scalarization. This solution simply consists in introducing a preference vector $\lambda$ that defines a trade-off between the $n$-conflicting objectives. Thereafter, proceed to optimize the weighted sum of the objectives as one.

$$\min_{\theta \in \mathbb{R}^d} \left( f(\theta) \stackrel{\text{def}}{=} \sum_{i=1}^{n} \lambda_i f_i(\theta) \right). \tag{2}$$

The vector $\lambda$ is referred to as the preference or trade-off vector and is typically chosen such that $\sum_{i=1}^{n} \lambda_i = 1$ and $\lambda_i > 0$.

Our insight is to treat the Byzantine-robust learning problem as multi-objective, then to deploy linear scalarization to solve it. That is, the trade-off vector can be leveraged to balance users' updates by penalizing the suspected Byzantine ones without dropping any user's contribution. Additionally, this scheme addresses the trade-off between granting Byzantine resilience and inclusion as no client update is fully discarded.

Our contribution can be summarized as follows:

- We propose $RAGG$-LS, a linear scalarization based aggregation that leverages existing defenses to define a trade-off vector over the $n$-conflicting objectives in order to grant Byzantine resilience in the non-IID setting. Moreover, the fact that the trade-off vector is defined through robust aggregations balances out updates according to the level of trust in the corresponding client.

- We evaluate our scheme against state of the art Byzantine attacks. We compare our approach to that of bucketing introduced in Karimireddy et al. (2022) as well as the standard IID defenses.

## 1.1 NOTATIONS AND SET UP

**Notations.** In the following $RAGG(.)$: Denotes a given Robust Aggregation Rule. $f_i$: The objective function associated with the $i$-th worker. $\mathcal{G} = \{g_1, g_2, ..., g_n\}$: corresponds to users gradient vectors. $\mathcal{G}_t$, $\mathcal{G}_b$ Respectively denote the sets of gradient vectors associated with the set of trusted workers and workers suspected to be Byzantine. $\lambda$: the preference or trade-off vector. $n$ the total number of clients, among which a subset $b$ behaves maliciously. $\delta$: The proportion of Byzantine clients, $\delta_{max} < \frac{n}{2}$. $\mathcal{D}_i = \{(x_j, y_j)\}_{j=1}^{|\mathcal{D}_i|}$ the subset of data of worker $i$. $C_{R_i}$ denotes a robust metric computed based on the gradient update of user $i$.

**Set Up.** We consider the standard Parameter Server (**PS**) architecture, comprised of a central server in charge of the exchange between the $n$ nodes. The (PS) architecture assumes that the server is trusted and a proportion $\delta$ of the nodes behave arbitrarily or maliciously. The arbitrary behavior is modelled following the Byzantine abstraction introduced in Lamport et al. (1982). The Byzantine workers maybe omniscient (i.e., they have access to the whole dataset and are able to access other clients' updates). We assume data distribution among clients to be non-IID. We are interested in studying the Byzantine Resilience of Distributed Synchronous SGD.

## 1.2 OUTLINE

We structure the paper as follows: In section 2 we present an overview of key relevant literature. We then introduce a formulation of our approach in section 3. The final two sections respectively present the experimental evaluation and concluding remarks.

## 2 RELATED WORK

In this section relevant work is highlighted. We first present some of the renowned aggregation rules in the IID then in the non-IID setting. Followed by Byzantine attacks on $SGD$. Finally, we present some of the works that deploy linear scalarization to balance trade-offs between the objectives.

**IID Defenses**: **Krum** Blanchard et al. (2017) is an aggregation that filters the gradient submissions based on a trust score defined as follows: $s(i) = \sum_{j \in \mathcal{N}_i} dist(g_i - g_j)^2$ where $\mathcal{N}_i$ denotes the set of $n - b - 2$ closest vectors of $i$. The aggregation selects the gradient vector of the user $i$ minimizing the score $s(i)$. The authors also introduce a variant namely $m$-**Krum** that returns an average of the $m$ vectors with the smallest scores. **The Median** Yin et al. (2018) consists in applying the median along each coordinate of the $d$-dimensional gradient vectors. **Trimmed-mean** Yin et al. (2018) averages the vectors submissions after discarding a proportion of the highest and smallest values. **MeaMed** operates by computing the mean around the median of vector updates. **Bulyan** El Mhamdi et al. (2018) is a meta gradient aggregation rule that grants strong Byzantine resilience (defends against adversaries exploiting the high dimensionality of the model). It is used as an enhancing method on top of a weak Byzantine aggregation. Bulyan operates in two steps: First, it uses the initial aggregation to get a set of trusted users. Then, it uses a variant of the trimmed-mean to aggregate the trusted set. **Aksel** Boussetta et al. (2021) combines the median and norm filtering to construct a robust interval that is used to filter the gradient vectors. The output of the filtering step is then averaged. Alistarh et al. (2018) uses MeaMed (the mean around the meadian) as a gradient aggregation rule (GAR) alongside an iterative scheme that allows the detection of Byzantine workers based on their submissions (meeting an upper bound on the error introduced on the submission in different rounds leads to the elimination of the node's contribution in subsequent steps). The work of Guerraoui (2021) offers a standard implementation of state of the art aggregations in a (PS) architecture under two of the powerful attacks against SGD Xie et al. (2019); Baruch et al. (2019) and show that introducing Nesterov's momentum at the workers improves the defenses' resilience. Similarly, Karimireddy et al. (2021) illustrate the role of momentum in strengthening the aggregations capacity of withstanding Byzantine attacks in the non-IID setting.

**Non-IID Defenses**: Li et al. (2018) propose a new aggregation RSA that uses an $l_1$ regularization term to penalize the deference between local iterates and server iterate. This approach however is unable to defend against state of the art Byzantine attacks (Baruch et al. (2019); Xie et al. (2019)). The work in Karimireddy et al. (2021) highlights the weakness of standard Byzantine defenses in the non-IID setting and proposes a Centered Clipping (CClip) method and the use of momentum to defend against time coupled attacks. Karimireddy et al. (2022) proposes a bucketing technique that reduces inter-client variance by averaging clients updates within buckets. This approach is simple and can be combined with existing defenses as a prepossessing step.

**Byzantine Attacks**: The Byzantine abstraction (Lamport et al. (1982)) encloses attacks ranging from random outputs, noisy submissions, sign flipping to carefully crafted updates targeting specific algorithms. In this we focus on malicious attacks that disrupt the convergence of $SGD$. A little is enough **ALIE** (Baruch et al. (2019)) is a non-omniscient attack wherein the adversaries estimate the mean $\mu$ and standard deviation $\sigma$ of the good gradients, and send $\mu - z\sigma$ to the server where $z$ is a small constant controlling the strength of the attack. **IPM** (Xie et al. (2019)) or Attack by Inner Product Manipulation is an omniscient attack (i.e., Byzantine workers are assumed to have access to the dataset). The attackers averages the gradients computed using the full dataset $Avg(G)$ then sends $-\epsilon Avg(G)$ where $\epsilon$ controls the strength of the attack. **Bit Flip (Sign Flip)** is a simple attack that models hardware failures in clients. A given Byzantine client experiences an internal failure and starts outputting faulty values in this case inverting the sign of the update. **Mimic** (Karimireddy et al. (2022)) is an attack that is designed to take advantage of the non-IID setting. The malicious workers pick an honest worker $i^*$ and mimic its output. This attack is powerful as it emphasises the heterogeneity by over-representing a worker $i^*$ with least significant update.

**Linear Scalarization**: is a technique used in Multi-Objective Optimization (MOO) to balance conflicting objectives. The formulation of leaning algorithms as multi-objective problems is indeed a more natural way of addressing them. In (Sener & Koltun (2018)) Multi-Task Learning (MTL) is modelled as a multi-objective problem, Lin et al. (2019) propose an efficient method, Pareto MTL, to solve Multi-Task Learning as MOO. Mohri et al. (2019) propose a reformulation of Federated Learning namely: Agnostic Federated Learning ($AFL$) to instate fairness among users through the optimization of the weighted sum of the users' objectives for the worst trade-off vector. However, we are not aware of any work addressing the Byzantine Learning problem as MOO. There is however a line of work that deploys weights or trust scores to penalize the Byzantine (Peng et al. (2020); Regatti & Gupta (2020); Fu et al. (2019); Li et al. (2018)). However, for the above-mentioned works lambda is generally used for regularization and not as a trade-off vector i.e., the problem is not solved as LS of conflicting objectives. Also, defining the trade-off is a crucial part in balancing the objectives. However, in the aforementioned works lambda is chosen arbitrarily or at best is not carefully tailored to each client in a way that would penalize the Byzantine.

## 3 ALGORITHM

### 3.1 STUDIED ROBUST AGGREGATIONS

Given a Robust Aggregation rule $RAGG(.)$, the Byzantine defense generally operates as follows: Apply a criterion (historical data, robust statistics, distance measure ...) to filter users into two subsets. Thereafter proceed to aggregate the gradients of trusted users $\mathcal{G}_t$. More precisely (i) Use some trust score or robust statistic $C_{R_i}$ to split users into two subsets honest $\mathcal{G}_t$ and suspected malicious $\mathcal{G}_b$. (ii) Return the aggregation of $\mathcal{G}_t$.

The proposed framework serves as an enhancements to existing robust defenses. We cover three aggregations namely: $m$-Krum, Aksel and the coordinate-wise median (CM). First, we explicitly define the procedure followed by each of these aggregations.

$m$-**Krum** Blanchard et al. (2017): For each user $i$ Krum computes a score $s_i$ for user $i$ based on the Euclidean distance of $g_i$ to $g_j$ such that $j \in \mathcal{N}_i$ which denotes the closest $n - b - 2$ users to $i$, $s(i) = \sum \|g_i - g_j\|^2$, then it selects the user with the smallest score, i.e,

$$Krum(g_1, g_2, ..., g_n) = \arg\min_i s(i).$$

The $m$-Krum variant picks $m$ users with the smallest scores then outputs their average.

**CM** Yin et al. (2018): The CM, uses the one-dimensional median across each dimension to filter out outliers. For $g_i \in R^d$, $CM(\mathcal{G}) = g_M$, where $g_M$ is a vector with its $k$-th coordinate being $g_M[k] = Median\{g_i[k] \mid i \in [m]\}$ for $k \in [d]$.

**Aksel** Boussetta et al. (2021) combines CM and norm filtering to select a subset of trusted gradients $\mathcal{G}_t$. The aggregation thereafter is the average of selected gradient vectors.

$$Aksel(\mathcal{G}) = \frac{1}{|\mathcal{G}_t|} \sum_{k=1}^{|\mathcal{G}_t|} g_k.$$

Where $g_M = CM(\mathcal{G})$. $S = \{s(i) = \sum_{k=1}^{d}(g_i[k] - g_M[k])^2 : i \in [n]\}$. $r = Median(S)$ and $I$ the robust interval $I = [0, r]$ and $\mathcal{G}_t = \{g_i \mid \|g_i - g_M\|^2 \in I\}$.

### 3.2 BYZANTINE LS APPROACH

Let $C_{R_i}$ denotes a criterion that measures the honesty of a client $i$. This is generally a vote/score (e.g., squared sum of Euclidean distance Blanchard et al. (2017) or the distance to some robust statistic). Naturally, the goal is to pick the gradient updates associated with the users having the smallest $C_{R_i}$. Following that, the main idea behind $LS$ is, rather than discarding the suspected users that stray from $C_{R_i}$, each user is penalized according to the degree of derail. As such, we associate with each user $i$ a scalar $\lambda_i$ namely a trade-off that is defined such that $\lambda_i \approx \frac{1}{C_{R_i}}$. In the following we summarize the main LS steps

1. Use $RAGG(.)$ to split users into two subsets honest $\mathcal{G}_t$ and suspected malicious $\mathcal{G}_b$.
2. Deploy the selection criterion to define the trade-off vector $\lambda$.
3. Return the weighted sum of all users gradients $RAGG\text{-}LS = \sum_i \lambda_i g_i$.

LS Leverages the standard aggregations to separate the gradients into honest and suspected Byzantine. Then, utilizes the filtering metric to define a trade-off vector such that the more suspected a user is to be Byzantine, the smaller its weight is. That is, our approach keeps all users' gradient vectors and the $\lambda_i$ serves as a penalty: as such, for $i$ in the subset of discarded gradients, the penalty associated with the re-introduction of $g_i$ is computed as follows $\lambda_i = \alpha_b \frac{1}{C_{R_i}}$. In this case $\alpha_b$ defines the trade-off between inclusion and robustness and quantifies the confidence in the $C_{R_i}$.

$$\alpha_t \sum_{i \in \mathcal{G}_t} \frac{1}{C_{R_i}} g_i + \alpha_b \sum_{i \in \mathcal{G}_b} \frac{1}{C_{R_i}} g_i \tag{3}$$

Thereafter, LS can be considered as a generalization to the standard $RAGG(.)$, that pick $\lambda_i = 0$ for the suspected Byzantine and associates uniform weights with the honest clients. Additionally, this formulation alleviates the Robustness/Inclusion trade-off. Algorithm 1 gives the pseudo code of our approach.

---

**Algorithm 1** RAGG-LS

---

    **Input** $\mathcal{G}$, two hyper parameters $\alpha_t > 0$ and $\alpha_b \geq 0$,
    **Compute** $C_{R_i}$ based on the chosen RAGG
    **Split** $\mathcal{G}$ into $\mathcal{G}_t$ (honest clients) and $\mathcal{G}_b$ (suspected malicous clients) based on the chosen RAGG.
    **for** $g_i \in \mathcal{G}_t$ **do** $\lambda_i = \frac{\alpha_t}{C_{R_i}}$
    **end for**
    **for** $g_i \in \mathcal{G}_b$ **do** $\lambda_i = \frac{\alpha_b}{C_{R_i}}$
    **end for**
    **Normalise** $\lambda$
    **Return** $RAGGLS(\mathcal{G}) = \sum_{i=1}^{|\lambda|} \lambda_i g_i$

---

## 4   Empirical Evaluation

In this section we evaluate the introduced LS approaches on non-IID data when a proportion of the clients is Byzantine. We first give a detailed description of the experimental set up, namely: The used data sets and the techniques deployed to partition the data between client in a non-IID fashion. Followed by the models' architectures alongside all the relevant hyper-parameters. We also define the threat model and the studied defenses. Finally, we present Top-1 Accuracy as an evaluation metric.

### 4.1   Description

**Data Splits**: We study the following datasets namely: MNIST LeCun et al. (1998), FMNIST Xiao et al. (2017), SVHN Netzer et al. (2011) and CIFAR10 Krizhevsky & Hinton (2009). We experiment with IID and non-IID data splits. The later is generated by creating label unbalances between clients.

A commonly used non-IID partition method, Latent Dirichlet Sampling. This split approach uses the Dirichlet distribution $Dir(\beta)$, such that each party gets assigned a proportion of samples of a given label. We consider $\beta$ a hyper-parameter that allows to introduce varying levels of skewness (the smaller the $\beta$ the more skewed is the resulting split). Following existing works Li et al. (2022); Tang et al. (2022), each dataset is partitioned with two different non-IID degrees using $\beta = 0.1$ for mild non-IID and $\beta = 0.01$ for high non-IID splits.

Moreover, to verify the impact of LS, we conduct experiments using another type of non-IID partition: $C_k$-classes partition, it is common to use $K = 1$ a split referred to as pathological non-IID McMahan et al. (2016), we however set $K = 3$ i.e. each client only has access to labels from 3 classes.

**Learning Model**:

- **Model**: $CNN$ with two convolutional layers, Max-Pooling and a Fully-connected layer of 1k units for MNIST and FMNIST, $LeNet5$ LeCun et al. (1998) for SVHN and $VGG11$ Simonyan & Zisserman (2014) for CIFAR10.
- **Loss function**: Cross Entropy Loss.
- **Optimizer**: SGD with $momentum = 0.9$ at the workers, a fixed learning rate 0.01 and a batch size of 128.

**Threat Model**: We experiment with two values of Byzantine $\delta = 20\%$ and $\delta = 40\%$ i.e., out of 25 users respectively 5 or 10 users are byzantine. The deployed attacks are indicated bellow:

- **ALIE** (Baruch et al., 2019): Following the original paper we use $z = 0.25$ computed using the formula from the original paper, this also the value used in Karimireddy et al. (2022).
- **IPM** (Xie et al., 2019): We use $\epsilon = 0.1$ following Karimireddy et al. (2022).
- **BF**: A Byzantine worker sends $-\nabla f_i$ instead of $\nabla f_i$ due to hardware failures etc.
- **Mimic**: The goal of the mimic attack as introduced in Karimireddy et al. (2022) is the over-representation of a user $i^*$ with the least significant gradient update, a method to pick $i^*$ is introduced, we however implement mimic in a simpler way. We utilize the entropy of users unbalanced data to get insight on the quality of each users update. Thereafter, Mimic picks the client $i^*$ such that,

$$i^* = \arg\min_i \; Entropy(\mathcal{D}_i).$$

**Defenses**: We explore the Byzantine resilience of standard defenses and compare them to their non-IID variants based on Baucketing and LS.

- **IID defenses** namely, CM, mKrum and Aksel.
- **Bucketing** ($s = 2$) CM-buck, mKrum-buck and Aksel-buck.
- **Linear scalarization variants**: We denote the LS variants of the previous methods as follows: CMLS 2 for LS variant of coordinate wise median, mKLS 3 for mKrum LS variant and ALS 4 for Aksel LS variant. We set the hyper parameters $\alpha_t = 1$ and $\alpha_b = \frac{1}{9}$, as such we keep the trade-off associated with honest workers as is computed and we further penalize the suspected Byzantine by multiplying their $\lambda_i$ by a small constant.

### 4.2 EXPERIMENTAL RESULTS

In this section we present the experimental results. In their work, Karimireddy et al. (2022) demonstrate the effectiveness of their approach on the MNIST dataset for a given non-IID split under a fixed number of Byzantine namely 20%. In the following we test the bucketing scheme alongside the introduced LS variants and the defenses they improve under Byzantine attacks (i) On varying degrees of non-IID to the data (ii) on multiple datasets namely: MNIST, CIFAR10 and SVHN. Additional experiments on both MNIST and FMNIST can be found in the Appendix fig.9, fig.10, fig.11

We report $Top-1$ Test Accuracy averaged over 5 runs for each $GAR$ for MNIST, 3 runs for SVHN and 2 runs for CIFAR10, under no attack (NA), ALIE, IPM, BF and Mimic for the IID data split and 2 different non-IID scenarios based on label unbalance between the users.

- Our initial results suggest that our approach is viable in the IID case and achieves results comparable to state of the art defenses under different Byzantine attacks fig. 1.
- In the mild non-IID case i.e., Data $\sim Dir(\beta = 0.1)$ and Data $\sim C_3$ , our approach as well as bucketing help the standard aggregations perform better in the non-IID Byzantine setting. Also in the mild non-IID setting our approaches are either comparable to bucketing or outperform it when the proportion of Byzantine $\delta = 20\%$ fig.2. However, when the proportion of byzantine is doubled i.e., $\delta = 40\%$ our variants are outperforming all the other aggregations including their corresponding bucketing variants fig.4.
- Finally, in the strongly non-IID case $Dir(\beta = 0.01)$ the LS based methods are the only ones capable of learning on heterogeneous data in the presence of Byzantine clients. fig.3.

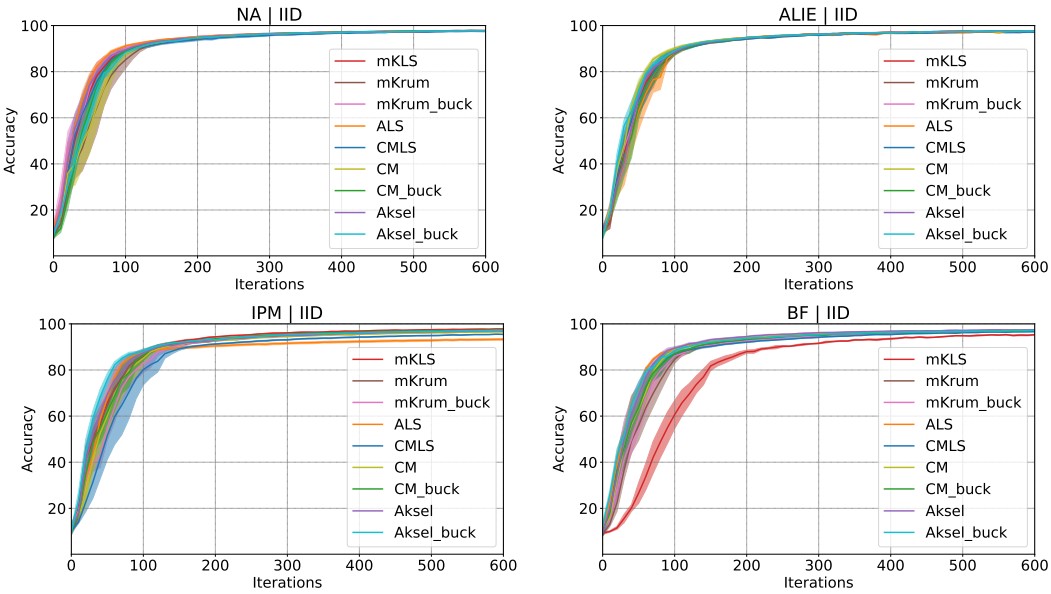

Figure 1: MNIST $Top-1$ Test Accuracy for all GARS on IID data split under the following attacks **NA**: No Attack, **ALIE**: A Little Is Enough, **IPM**: Inner Product Manipulation and **BF** Bit Flip attack

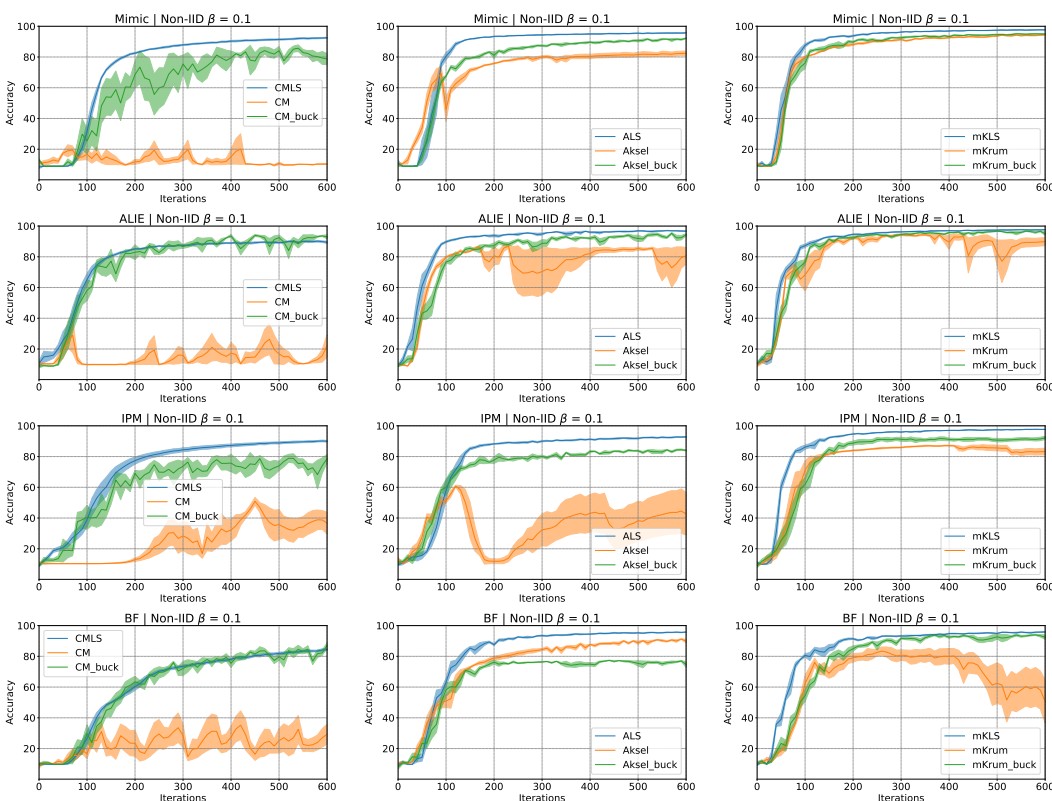

Figure 2: MNIST $Top-1$ Test Accuracy for All GARs on Non-IID data $\sim Dir(\beta = 0.1)$ (We consider this mild non-IID split, all users have access to varying proportions of the dataset labels). Each plot compares a standard $RAGG$ to its bucketing variant and its LS variant. Each row corresponds to a given Byzantine attack, from the left respectively we have: **Mimic**: mimic Attack, **ALIE**: A Little Is Enough, **IPM**: Inner Product Manipulation and **BF** Bit Flip attack

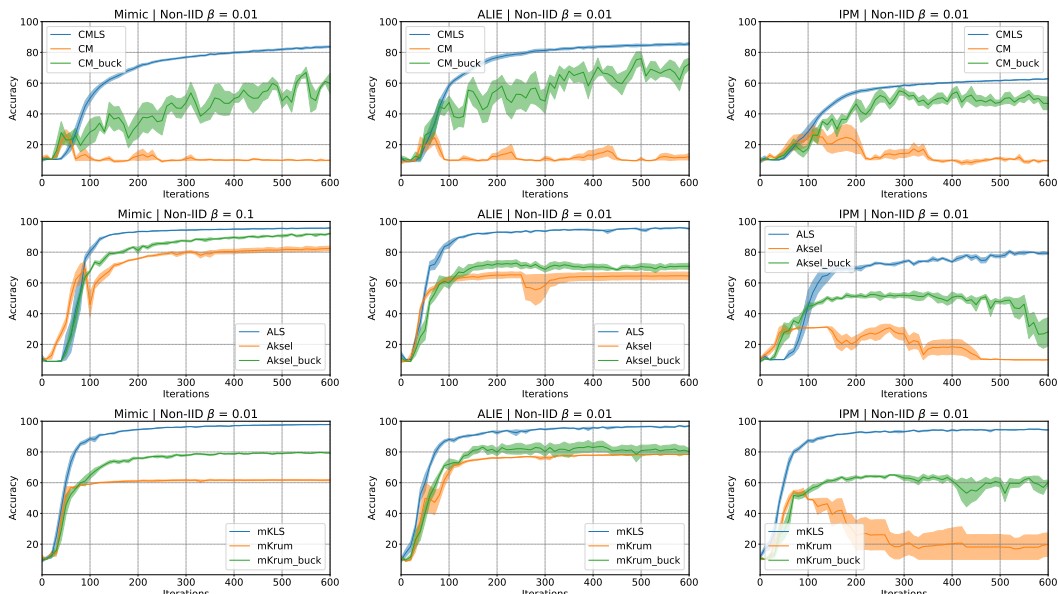

Figure 3: MNIST $Top-1$ Test Accuracy for All GARs on Non-IID data $\sim Dir(\beta = 0.01)$ (We consider this High non-IID split, all users have access to varying proportions of the dataset labels). Each plot compares a standard $RAGG$ to its bucketing variant and its LS variant. Each column corresponds to a given Byzantine attack, from the left respectively we have: **Mimic**: mimic Attack, **ALIE**: A Little Is Enough, **IPM**: Inner Product Manipulation

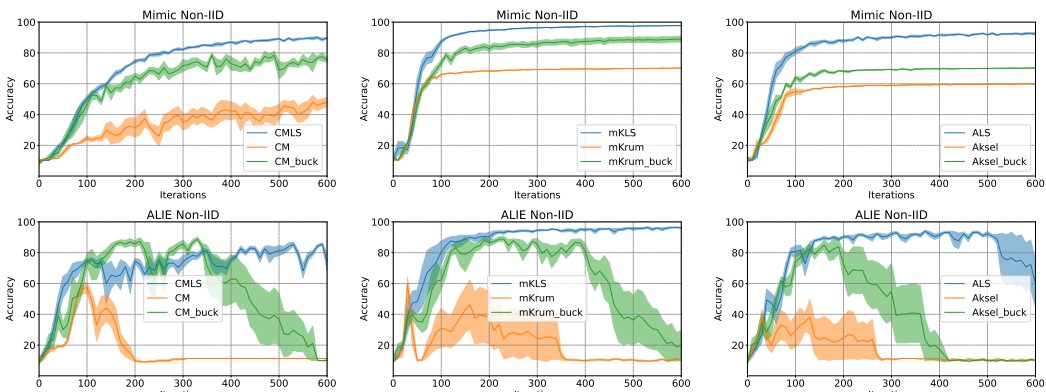

Figure 4: MNIST $Top-1$ Test Accuracy for All GARs on Non-IID $C_3$ under $\delta = 40\%$ Byzantine clients. Each plot compares a standard $RAGG$ to its bucketing variant and its LS variant under **Mimic** and **ALIE** attack

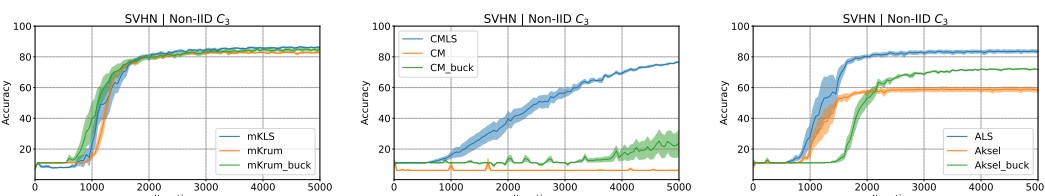

Figure 5: **SVHN** $Top-1$ Test Accuracy for All GARs on Non-IID $C_3$ under $\delta = 20\%$ Byzantine clients. Each plot compares a standard $RAGG$ to its bucketing variant and its LS variant under **Mimic** attack

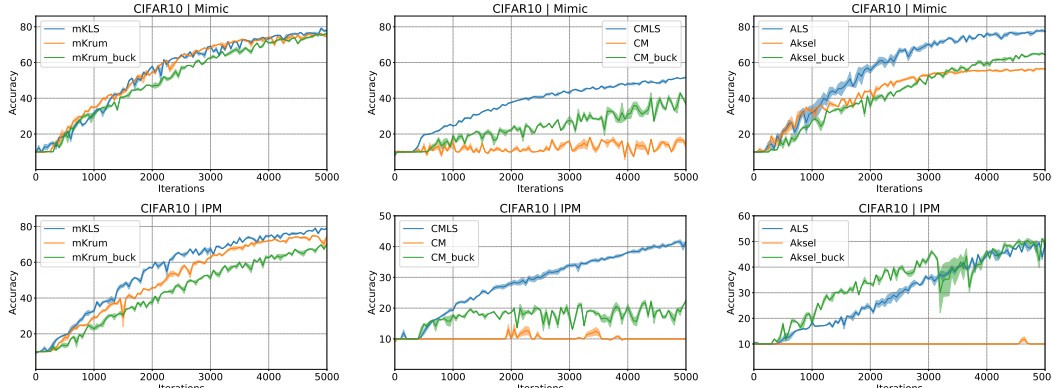

Figure 6: **CIFAR10** $Top-1$ Test Accuracy for All GARs on Non-IID $C_3$ under $\delta = 20\%$ Byzantine clients. Each plot compares a standard $RAGG$ to its bucketing variant and its LS variant under **Mimic** and **IPM** attack

Although bucketing does improve the standard aggregations in some scenarios mainly mild non-IID under 20% Byzantine, however as data distribution gets highly unbalanced fig.3 or the proportion of Byzantine augments fig.4 it is no longer efficient in alleviating the impact of heterogeneity on the performance of the aggregations.

On CIFAR10 fig.6 and SVHN fig.5 the LS variants outperform all the other aggregations under 20% Byzantine attackers (Mimic attack). On the MNIST dataset, our approaches are either comparable to bucketing or outperform them on the mild non-IID case under 20% of Byzantine. However, when the number of Byzantine is doubled or non-IID degree is high our variants are the only ones that achieve reasonable accuracy under Byzantine attacks.

## 5 CONCLUSION

In this work we explore the problem of learning on heterogeneous data in the presence of Byzantine clients. We propose a Linear Scalarization approach that takes advantage of the existing Byzantine defenses to define a trade-off between the client's objectives. This formulation allows to balance users' contribution in a way that the honest workers' gradient vectors are of great importance and those of suspected workers have a small contribution. Our approach grants Byzantine robustness while allowing outliers' contributions. Thereafter alleviating the Inclusion/Robustness trade-off posed by standard aggregations.

Our LS methods are a generalization of the standard $RAGG(.)$, that pick $\lambda_i = 0$ for all suspected malicious clients. Thus, the theoretical guarantees of these methods extend to the particular setting of our methods especially for IID data. However in non-IID setting, our work still lacks on the theoretical side, this is left for future work. Another limitation of our work is the reliance on standard aggregation to compute the trade-off vector. Although the rationale is to make sure the trade-off is based a Byzantine-robust trust score / metric, it however is tied to the computational complexity of the deployed aggregation that in cases like m-Krum is $O(n^2 d)$ where $d$ is the dimensionality of the model. Thus, it would be interesting to investigate more efficient ways of defining suitable trade-offs between the clients.

Additional future improvements may address the tuning of the hyper-parameters $\alpha_t$ and $\alpha_b$ and the communication efficiency as it is one of the challenges in FL.

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

# A  APPENDIX

## A.1  ALGORITHMS

In this section we present the algorithms corresponding to the introduced LS variants.

---

**Algorithm 2** CMLS

---

$g_M = CM(\mathcal{G})$ and $\lambda_M = \alpha_t$         ▷ This is considered the only entry in $\mathcal{G}_t$
**for** $g_i, i \in [n]$ **do**
    $\lambda_i = \frac{\alpha_b}{\|g_i - g_M\|}$      ▷ Defining the trade-off vector associated with users gradients
**end for**
$\lambda = (\lambda_M, \lambda_1, ..., \lambda_n)$            ▷ Normalise $\lambda$
**Return** $CMLS(\mathcal{G}) = \sum_{i=1}^{n+1} \lambda_i g_i$

---

---

**Algorithm 3** MKLS

---

$\mathcal{G}_t$ contains the selected vectors using $m$-Krum.
**for** $g_i, i \in [n]$ **do**
    **if** $g_i \in \mathcal{G}_t$ **then**         ▷ Defining the trade-off vector using $m$-Krum scores $s_i$
        $\lambda_i = \frac{\alpha_t}{s_i}$
    **else if** $g_i \in \mathcal{G}_b$ **then**
        $\lambda_i = \frac{\alpha_b}{s_i}$
    **end if**
**end for**
$\lambda = (\lambda_1, ..., \lambda_n)$            ▷ Normalise $\lambda$
**Return** $MKLS(\mathcal{G}) = \sum_{i=1}^{n} \lambda_i g_i$

---

---

**Algorithm 4** ALS

---

$\mathcal{G}_t$ contains the selected vectors using Aksel.
**for** $g_i, i \in [n]$ **do**
    **if** $g_i \in \mathcal{G}_t$ **then**         ▷ Defining the trade-off vector using Aksel
        $\lambda_i = \frac{\alpha_t}{\|g_i - g_M\|^2}$
    **else if** $g_i \in \mathcal{G}_b$ **then**
        $\lambda_i = \frac{\alpha_b}{\|g_i - g_M\|}$
    **end if**
**end for**
$\lambda = (\lambda_1, ..., \lambda_n)$            ▷ Normalise $\lambda$
**Return** $ALS(\mathcal{G}) = \sum_{i=1}^{n} \lambda_i g_i$

---

# B  EMPIRICAL EVALUATION

## B.1  DESCRIPTION

**1. Hardware and software specifications** Our codes are implemented using PyTorch 1.9.0 with cuda 11.1. Experiments are run on the GPU partition of a cluster on a single node containing:

- 2x AMD EPYC 7713 64-Core Processor 1.9GHz (128 cores in total)
- 1000 GiB RAM
- 4x NVIDIA A100-SXM4-80GB GPUs
- Dual-rail Mellanox HDR200 InfiniBand interconnect

## B.2 EXPERIMENTAL RESULTS

In this section we present additional experimental results on both MNIST and FMNIST dataset.

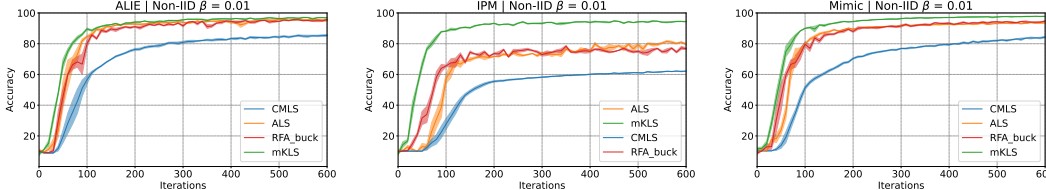

Figure 7: MNIST $Top - 1$ Test Accuracy for the LS variants compared to RFA with bucketing (RFA_buck) on non-IID $\sim Dir(\beta = 0.01)$ split. Each plot compares RFA with bucketing to the LS variants. Each row corresponds to a given Byzantine attack, from the left respectively we have: **Mimic**: mimic Attack, **ALIE**: A Little Is Enough, **IPM**: Inner Product Manipulation

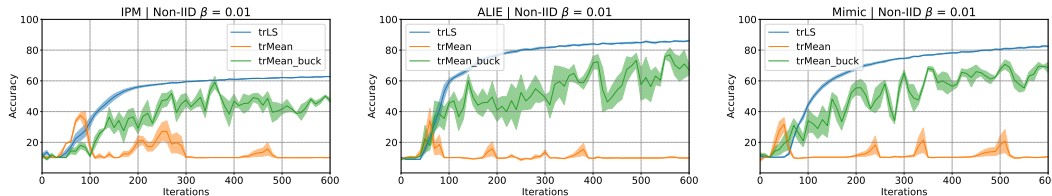

Figure 8: MNIST $Top - 1$ Test Accuracy for trMean to trLS and trMean_buck, its LS and bucketing variants on non-IID $\sim Dir(\beta = 0.01)$ split. Each row corresponds to a given Byzantine attack, from the left respectively we have: **Mimic**: mimic Attack, **ALIE**: A Little Is Enough, **IPM**: Inner Product Manipulation

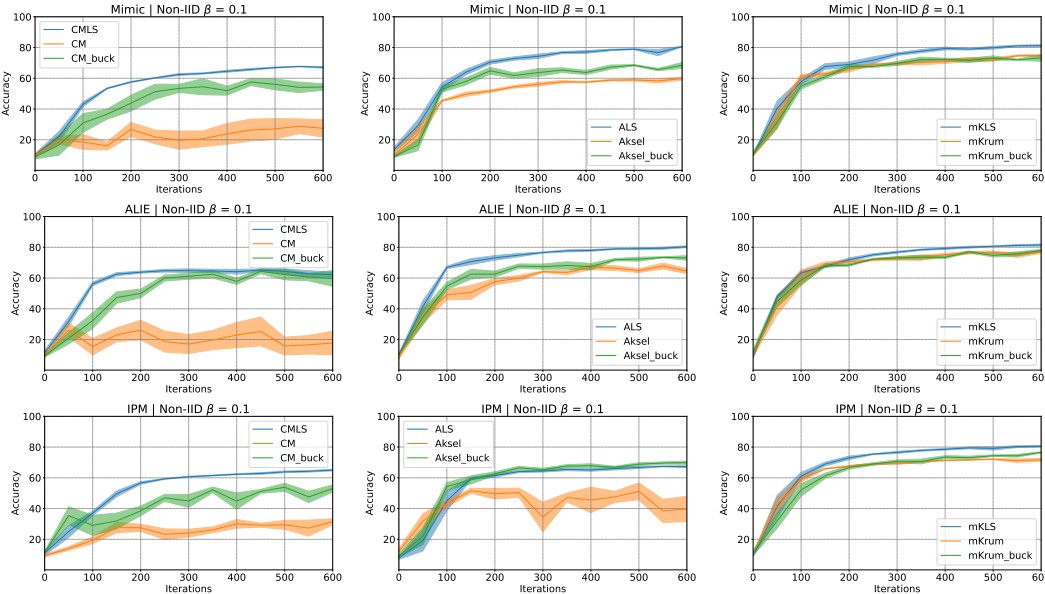

Figure 9: FMNIST $Top - 1$ Test Accuracy for all GARS on non-IID $\sim Dir(\beta = 0.1)$ split. Each plot compares a standard $RAGG$ to its bucketing variant and its LS variant. Each row corresponds to a given Byzantine attack, from the left respectively we have: **Mimic**: mimic Attack, **ALIE**: A Little Is Enough, **IPM**: Inner Product Manipulation

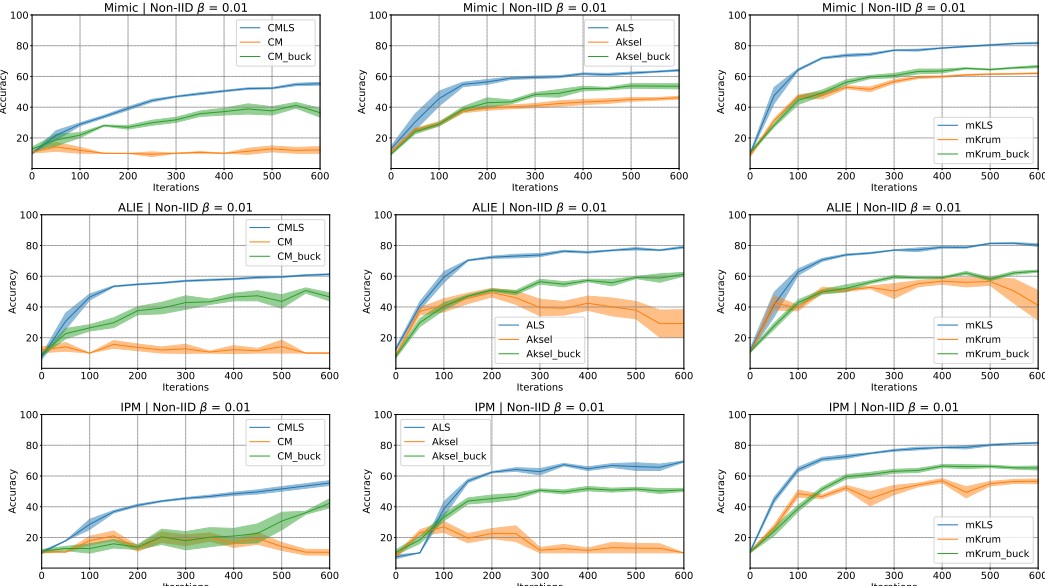

Figure 10: FMNIST $Top - 1$ Test Accuracy for all GARS on non-IID $\sim Dir(\beta = 0.01)$ split. Each plot compares a standard $RAGG$ to its bucketing variant and its LS variant. Each row corresponds to a given Byzantine attack, from the left respectively we have: **Mimic**: mimic Attack, **ALIE**: A Little Is Enough, **IPM**: Inner Product Manipulation

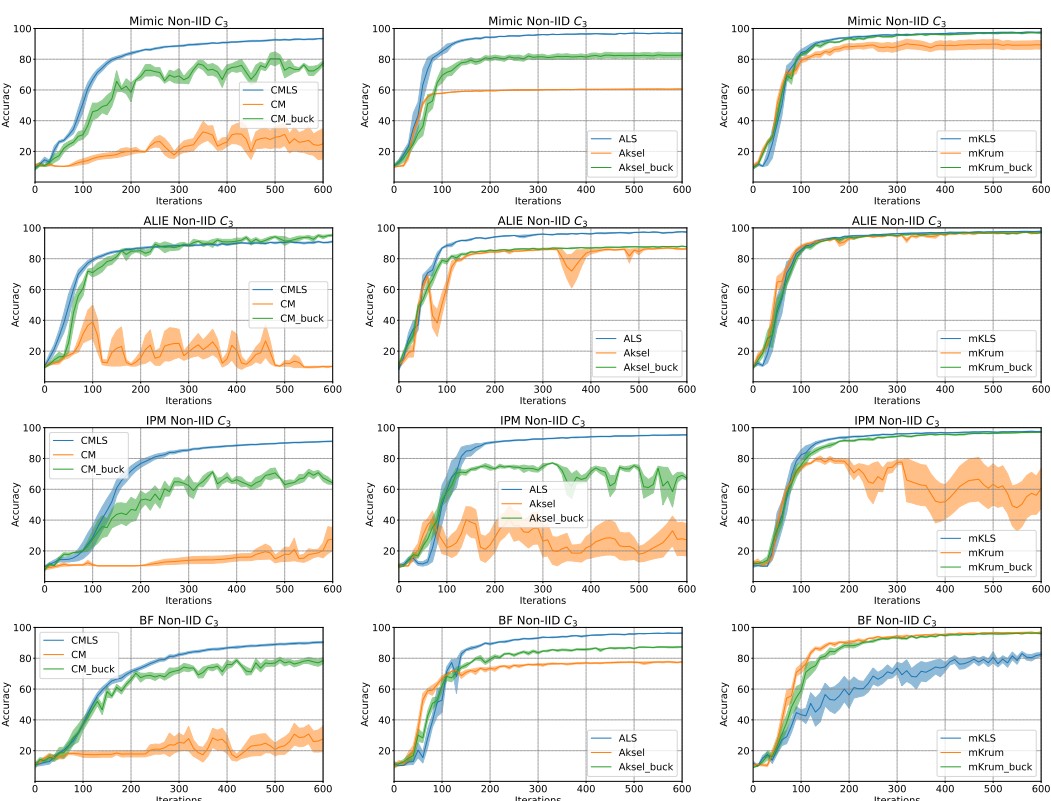

Figure 11: MNIST $Top - 1$ Test Accuracy for all GARS on Non-IID $C3$ split (i.e. each agent only has access to samples from 3 labels). Each plot compares a standard $RAGG$ to its bucketing variant and its LS variant. Each row corresponds to a given Byzantine attack, from the left respectively we have: **Mimic**: mimic Attack, **ALIE**: A Little Is Enough, **IPM**: Inner Product Manipulation and **BF** Bit Flip attack

