# OpenReview forum: "Linear Scalarization for Byzantine-Robust Learning on non-IID data"
_ICLR.cc/2023/Conference — Submitted to ICLR 2023_

### Official Review · Reviewer_xe7w · 2022-10-23

**Confidence:** 4
**Correctness:** 2
**Technical Novelty And Significance:** 2
**Empirical Novelty And Significance:** 2
**Recommendation:** 3

**Clarity, Quality, Novelty And Reproducibility:**

The readability of this paper is not good and many claims are not well-supported. Please see the concerns listed above for details.

**Strength And Weaknesses:**

The proposed method, called linear scalarization (LS), does not introduce too much computation cost and is easy to implement. However, there are also some concerns as follows.

1. The proposed method lacks enough theoretical support. LS is based on the existing aggregators RAGG, which are not robust on non-IID data, as mentioned in the paper. The split based on the chosen RAGG may be unreliable. Why the method based on a not robust aggregator is robust? It lacks theoretical support or even informal discussion about this.

2. There are two extra hyper-parameters $\alpha_t$ and $\alpha_b$ in the proposed method. However, there is not an adequate discussion on how to properly set the hyper-parameters. Meanwhile, the effect of the two hyper-parameters is not adequately studied in this work.

3. It is reported in [1] that bucketing with centered-clipping (CClip) and robust federated aggregation (RFA) has much better empirical performance. Given this, the comparison in this paper seems unfair.

4. In equation (2) (page 2), does it require that $\lambda_i \geq 0$? An explicit explanation is required here.

5. What is the definition of $\delta_{max}$ (page 2)? Is it the upper bound of $\delta$? Moreover, since there are $n$ clients in total, does $\delta<\frac{1}{2n}$ mean that there is no Byzantine client?

6. At the end of the paragraph about IID defenses (page 3), 'in the non-IID setting' --> ' in the IID setting'.

7. In the definition of CM (page 4), $M$ is defined to be an index while in the definition of Aksel, $M$ is defined to be a vector. The inconsistency will make readers confused. Meanwhile, it is suggested to represent vectors with bold letters, in order to avoid confusion.

8. What is the meaning of the abbreviation 'MOO' (the first paragraph of page 4)? Does it mean 'multi-objective optimization'? Readers can be confused by this.

9. There exist some typos and improper citation format. For example, there are several missing parentheses around the references in the paragraph about IID defense (page 2), in Section 3 (page 4), and in Section 4.1 (page 5). In the definition of Aksel (page 4), 'Combines' --> 'combines'.

[1] Karimireddy, Sai Praneeth, Lie He, and Martin Jaggi. "Byzantine-robust learning on heterogeneous datasets via bucketing." arXiv preprint arXiv:2006.09365 (2020).

**Summary Of The Paper:**

In this work, the authors study Byzantine-robust distributed learning on heterogeneous data and propose a new method called linear scalarization, where derailed clients are penalized via a trade-off vector. The proposed method is empirically compared with existing methods on several datasets.

**Summary Of The Review:**

This paper is not well-written. Meanwhile, the claims lack theoretical support and the empirical comparison is not solid enough. Given the reasons, there is much room for improvement and this work is currently below the acceptance threshold.

---

> ### Author Response · Authors · 2022-11-15
> **Author's response to  Reviewer xe7w**
>
> We thank the reviewer for providing valuable feedback to improve the work. In the following we try to address some of the reviewers concerns:
>
> *  An aggregation may discard benign updates due to the non-IID bias; LS allows the re-introduction of discarded updates with a penalty (the penalty serves as a precaution in the case that the discarded updates are indeed malicious). As such, LS can be seen as a corrective mechanism. An ideal formulation of LS however needs a $C_{R_i}$ independent of existing aggregation to define the trade-off (this point was mentioned in the limitation and future work part of the paper).
>
> * While we acknowledge the necessity of trying different values for $\alpha_t$, $\alpha_b$. It however is necessary to pick them such that the resulting trade-off is smaller for the suspected workers and bigger for the honest ones.
>     For $i$ in the subset of discarded gradients, the penalty associated with the re-introduction of $g_i$ is computed as follows $\lambda_i = \alpha_b \frac{1}{C_{R_i}}$. In this case $\alpha_b$ defines the trade-off between inclusion and robustness and quantifies the confidence in the $C_{R_i}$.  Eq (2) can be re-written as: $\alpha_t$ $\sum_{i \in \mathcal{G}t} \frac{1}{C{R_i}} g_i$ \+ $\alpha_b$ $\sum_{i \in \mathcal{G}b} \frac{1}{C{R_i}} g_i$. In that sense, as $\alpha_b$ gets smaller the contribution of suspected clients is milder.
>
> * We agree with the reviewer that [1] states the superiority of CCLIP alongside RFA with bucketing. The goal of our work was to enhance the remaining GARs that still degrade even with bucketing. Nonetheless, we add to the appendix a comparison  of RFA with bucketing to the LS variants see fig.7. (RFA with bucketing has comparable performance to ALS and is surpassed by mKLS).
>
> * We updated the paper and edited the inconsistencies and spelling.

---

### Official Review · Reviewer_hCuX · 2022-10-24

**Confidence:** 4
**Correctness:** 3
**Technical Novelty And Significance:** 3
**Empirical Novelty And Significance:** 1
**Recommendation:** 3

**Clarity, Quality, Novelty And Reproducibility:**

The paper needs to be further polished. There are still important issues in the proposed defense that need to be addressed. The authors do not provide code for reproducibility.

**Strength And Weaknesses:**

strength:

* The idea of trading off between granting Byzantine resilience and inclusion when the data is heterogeneous is interesting.

weaknesses:

* Proposed LS is not applicable to all RAGGs. Many RAGGs do not split gradients into honest gradients and suspected malicious gradients, e.g., trimmed mean [1], and geometric median [2]. How can LS enhance these methods?

* The selection criterion $C_{R_i}$ is important for LS. Elaborate more on how to choose an appropriate $C_{R_i}$.

* Incomplete experiment results. The performance against attack BF on MNIST under $\beta=0.01$ is missing (Figure 3); the performance against attack BF on MNIST under $\delta=40\%$ is missing; the performance against attacks BF, IPM, and ALIE on SVHN is missing; the performance against attacks BF and ALIE on CIFAR-10 is missing,

* Experiments on different $\alpha_t$s and $\alpha_b$s are expected. I am particularly interested in the case where $\alpha_t=\alpha_b$.

* I am wondering about the effect of applying LS to a particular defense DnC [3]

* The introduction of different RAGGs in Section 3.1 seems unnecessary.

[1] Yin, Dong, et al. "Byzantine-robust distributed learning: Towards optimal statistical rates." *International Conference on Machine Learning*. PMLR, 2018.

[2] Pillutla, Krishna, Sham M. Kakade, and Zaid Harchaoui. "Robust aggregation for federated learning." *arXiv preprint arXiv:1912.13445* (2019).

[3] Shejwalkar, Virat, and Amir Houmansadr. "Manipulating the byzantine: Optimizing model poisoning attacks and defenses for federated learning." *NDSS*. 2021.

**Summary Of The Paper:**

This paper focuses on Byzantine-robust learning when the data is heterogeneous. The authors propose a novel Linear Scalarization (LS). LS first uses RAGG to split gradients into honest gradients and suspected malicious gradients. Then, LS uses a selection criterion to compute weights for all gradients. In particular, LS assigns higher (lower) weights for honest (suspected malicious) gradients. Finally, LS computes the weighted average as the aggregated gradient.

**Summary Of The Review:**

The idea of trading off between granting Byzantine resilience and inclusion when the data is heterogeneous is interesting. However, some details of the proposed method need to be specified, and further experiments are needed to validate the efficacy of LS. Therefore, I think the paper is below the bar of ICLR.

---

> ### Author Response · Authors · 2022-11-15
> **Author's response to Reviewer hCuX**
>
> We thank the reviewer for providing valuable feedback to improve the work. In the following we try to address some of the reviewers concerns:
>
> * Although not all aggregations discard full gradient updates, however it holds that most aggregations discard a subset of information. In the case of $\beta$tr-Mean the aggregation discards $2 \beta n$ coordinates along each dimension. Although the resulting vector is blended from all the gradients outlier clients may still be discarded along coordinate.
>     The LS adaptation of tr-mean can be defined by assigning $C_{R_i}$ to be $\Vert g_i – trMean(g_1, g_2, …, g_n) \Vert$. We run few experiments (because of time limit) to test the LS adaptation of trMean and initial results show the improvement LS brings to trMean see fig.8 in the appendix.
>
> * Basically, a good $C_{R_i}$ is one that allows to quantify the honesty of a gradient submission taking as a reference the underlying GAR. However,  the ideal would be for LS to deploy an independent mechanism to define the $C_{R_i}$.
>
> * While we acknowledge the necessity of trying different values for $\alpha_t$, $\alpha_b$. It however is necessary to pick them such that the resulting trade-off is smaller for the suspected workers and bigger for the honest. For $i$ in the subset of discarded gradients, the penalty associated with the re-introduction of $g_i$ is computed as follows $\lambda_i = \alpha_b \frac{1}{C_{R_i}}$. In this case $\alpha_b$ defines the trade-off between inclusion and robustness and quantifies the confidence in the $C_{R_i}$. Eq (2) can be re-written as: $\alpha_t$ $\sum_{i \in \mathcal{G}t} \frac{1}{C{R_i}} g_i$ \+ $\alpha_b$ $\sum_{i \in \mathcal{G}b} \frac{1}{C{R_i}} g_i$.
>
>     In that sense, as $\alpha_b$ gets smaller the contribution of suspected clients is milder.
> * The adaptation of DnC with LS is straightforward as the GAR computes scores $s_i$ and discards outliers as a mechanism to categorise clients. One can set $C_{R_i} = s_i$. Due to the time limit on the period of editing and since we did not find any public implementation of DnC we were not able to run experiments to see how well the GAR performs with LS.

---

### Official Review · Reviewer_ZZGo · 2022-10-28

**Confidence:** 3
**Correctness:** 3
**Technical Novelty And Significance:** 2
**Empirical Novelty And Significance:** 2
**Recommendation:** 3

**Clarity, Quality, Novelty And Reproducibility:**

- The paper is well written and is of good quality. The novelty of this work is limited because of its incremental nature.
- Since there is no link to a public repo provided in the paper, this reviewer cannot comment on the reproducibility.

**Strength And Weaknesses:**

**Strengths**

- The results reported in the paper highlight the usefulness of the proposed method.

**Weaknesses**

- The paper has an incremental nature and the main contribution is minor in nature, which involves taking a convex linear combination of the gradients. There are no theoretical guarantees for the proposed algorithm and all the aggregation rules discussed in the paper have already been studied in the literature, along with rigorous guarantees. The connections to multi-objective optimization are also tenuous.
- Some of the claims made in the paper are either incorrect or misleading. E.g., "the main Byzantine defenses rely on the IID assumption causing them to fail when data distribution is non-IID even with no attack" is an incorrect statement (see, e.g., the supplementary material in BRIDGE: Byzantine-Resilient Decentralized Gradient Descent). Similarly, "Another weakness of current aggregations is that they all aim at totally discarding outliers for the sake of granting resilience." is a misleading statement. In the case of coordinate-wise trimmed mean, e.g., the final gradient typically gathers information from all the clients. And in the case of other aggregation rules, even when a client is excluded in one iteration, it can come back into the calculations in another iteration. The point of this discussion is not to dispute the idea that a better aggregation rule, such as the one proposed in this paper, might lead to better outcomes. The purpose is, however, to emphasize that the authors need to think a bit harder about their claims and reframe their contributions in light of these facts.

**Summary Of The Paper:**

The focus of this paper is on Byzantine-robust distributed learning. The authors in particular emphasize the setup where there is non-iid data across different clients. The main contribution in the paper is an algorithm in which the gradients from the clients are combined using a convex linear combination where the coefficients in the combination come from one of the robust aggregation rules that exist in the literature. The paper then provides several numerical results to showcase the benefits of the proposed algorithm.

**Summary Of The Review:**

While the paper is studying an important problem and the approach taken in the paper seems to be effective, the authors should build further on this work. Some more insights into the strength of this work as well as theoretical contributions would make this a stronger paper.

---

> ### Author Response · Authors · 2022-11-15
> **Author's response to Reviewer ZZGo**
>
> We thank the reviewer for providing valuable feedback to improve the work. In the following we try to address some of the reviewers concerns:
>
> * The claim concerning the limitation of the main aggregation rules in the non-IID setting is referenced from the results of the works of Karimireddy et al. [1] section 3.1. We clearly state in the introduction that the works of Karimireddy highlight the aforementioned result. Moreover, since in this work we considered the PS architecture, we only addressed the aggregations introduced in this set up, BRIDGE however works in the decentralized regime.
>
> * We agree that the claim "aggregations aim at totally discarding outliers" needs to be edited. We mean that in order to grant Byzantine-robustness 'the majority' of aggregations filter out some information (either complete gradient vectors or discard proportions of coordinates). In that case if a client is an outlier, it may be excluded from most rounds of training. Although not all aggregations discard full gradient updates, however it holds that most aggregations (at least the ones extensively studied in the literature) discard a subset of information: following the example on $\beta$trimmed-Mean (tr-Mean), this aggregation discards $2 \beta n$ coordinates along each dimension, where $\beta$ designates the proportion to drop. Although the resulting vector is blended from all the gradients, outlier clients may still be discarded along multiple coordinates especially in the case where data between clients is non-IID.
>
>
> [1] Karimireddy, Sai Praneeth, Lie He, and Martin Jaggi. "Byzantine-robust learning on heterogeneous datasets via bucketing." arXiv preprint arXiv:2006.09365 (2020).

---

### Official Review · Reviewer_TUza · 2022-10-31

**Confidence:** 4
**Clarity, Quality, Novelty And Reproducibility:** This paper is mostly clear and novel …
**Correctness:** 2
**Technical Novelty And Significance:** 3
**Empirical Novelty And Significance:** 3
**Recommendation:** 5

**Strength And Weaknesses:**

Strength:
- Good empirical performance.

Weakness:
- I am not convinced that LS addresses the essence of the non-iid issue, although it may reduce the influence of non-IID data distribution. Take CM+LS (algorithm 2) in the appendix for example. If there is no Byzantine workers and 49% of gradients are 0 while 51% of gradients are 1, then the output of CM+LS will be the same as CM which is 1. However, an ideal aggregation is expected to output something close to 0.5.

- There is no convergence guarantee as has been mentioned in the paper. A theoretical convergence guarantee is quite important because Byzantine robustness is designed to defend for all possible attacks. On the other hand, empirical evaluation only demonstrates its performance on a few attacks ---- there may exist some attacks that are designed to attack LS.

- Not clear how the number of tolerated Byzantine nodes changes with LS technique.



**Summary Of The Paper:**

This paper considers federated learning over non-iid data in the presence of Byzantine nodes. The authors propose to use a technique linear scalarizatoin (LS) to help other aggregators to achieve better performance. For example, instead of applying coordinate-wise median (CM) directly,  the CM+LS algorithm computes an aggregation weight for each gradient inversely proportional to their distance to the median. In this way, LS does not discard gradients but gives higher weight to good gradients. The empirical performance looks very nice but no theoretical guarantee is provided.

**Summary Of The Review:**

I would recommend borderline rejection because it does not address the main problem studied in the paper (non-iid) and there is no convergence guarantee.

---

> ### Author Response · Authors · 2022-11-15
> **Author's responde to Reviewer TUza**
>
> We thank the reviewer for providing valuable feedback to improve the work. In the following we try to address some of the reviewers concerns:
>
> * We do not claim addressing the essence of the non-IID problem, rather that LS enhances existing methods when data in non-IID (i.e., LS acts as a corrective measure: The underlying aggregation may filter out non-malicious users as a result of non-iid bias).
> The main goal of this work was to answer the following question:
> (Q) Can we grant Byzantine-robustness on non-IID data and inclusion of outliers without compromising the model’s accuracy ? In other words, can we get away with tolerating the outliers (Byzantine)?
>  Since the problem involves conflicting objectives then it may be solved through linear scalarization. The challenge however is how to define the trade-off vector. For that we resort to existing aggregations to define $C_{R_i}$.
> We acknowledge  that the fact that LS relies on the underlying GAR to compute the trade-off (penalty) is a limiting factor of the LS variants. This was mentioned in the future work section, we agree there needs to be a mechanism, independent of current aggregations, to define a suitable trade-off between the clients in a Byzantine non-IID learning set-up.
>
> * Concerning the theoretical analysis, we tried few ideas but for now we do not have enough strong results to present to the community. So we decided to continue investigating further and in the main time to differ it to a future work.

---

### Decision · Program_Chairs · 2023-01-20

**Decision:**

Reject

**Justification For Why Not Higher Score:**

Main concerns remained from all reviewers

**Justification For Why Not Lower Score:**

N/A

**Metareview: Summary, Strengths And Weaknesses:**

The paper studies Byzantine-robust distributed learning. It proposes to use existing gradient aggregation rules to derive weights for each contribution, aiming to give higher weights to trusted collaborators.

Unfortunately concerns remained from the reviews both on fundamentals (such as the fact that no convergence guarantees are given) and experiments, as well as the applicability to the important case of heterogeneous data.

We hope the detailed feedback helps to strengthen the paper for a future occasion.